# Enhanced Toughness and Sound Absorption Performance of Bio-Aerogel via Incorporation of Elastomer

**DOI:** 10.3390/polym14071344

**Published:** 2022-03-26

**Authors:** Junshi Shen, Ruofei Hu, Xueliang Jiang, Feng You, Chu Yao, Huan Yang, Peng Yu

**Affiliations:** 1Hubei Key Laboratory of Plasma Chemistry and Advanced Materials, School of Materials Science and Engineering, Wuhan Institute of Technology, Wuhan 430205, China; shenjunshi9707@163.com (J.S.); youfeng.mse@wit.edu.cn (F.Y.); chuyao@wit.edu.cn (C.Y.); yangh@wit.edu.cn (H.Y.); 2Department of Food Science & Chemical Engineering, Hubei University of Arts and Science, Xiangyang 441053, China; 3Hubei Key Laboratory of Polymer Materials, Hubei University, Wuhan 430062, China

**Keywords:** aerogels, Arabic gum, carboxylic butadiene-acrylonitrile rubber, mechanical performance, sound absorption

## Abstract

In this study, Arabic gum/ carboxylic butadiene-acrylonitrite latex aerogels (AG/XNBRL) hybrid aerogel was successfully prepared by a green method, i.e., the combination of latex compounding and vacuum freeze-drying process. After that, the obtained composites were subjected to a high temperature treatment to crosslink the rubber phase. It was found that the AG in the AG/XNBRL hybrid aerogel could act as a framework to improve the dimensional stability of the aerogel, while the XNBRL phase could significantly improve the mechanical flexibility of the ensuing composite. Compared to the AG aerogel which is highly brittle in nature, the AG/XNBRL hybrid aerogel not only exhibits significantly enhanced toughness, but also shows improved thermal stability and sound absorption performances; for instance, the half weight loss (50%) temperature and average sound adsorption coefficient for aerogel containing 30 wt% XNBRL is 344 °C and 0.585, respectively, which are superior to those of neat AG aerogel. Overall, this work provides novel inspiration to prepare the mechanical robust bio-based aerogel for the sound absorption application.

## 1. Introduction

With the fast urbanization and rapid growth of transportation, unwanted/excessive sound could have adverse effects on the work efficiency, living standards, and health of human beings [1]. Thereby, reducing noise by using sound absorption materials is of great importance to abate noise pollution [2,3]. Porous materials such as foams or aerogels have many channels so that sound waves are able to enter through them, and the porous materials are widely used to control noise as they can absorb most of the sound energy striking them. Most commercial foams, such as polyurethane foam, melamine foam, etc., are derived from non-renewable fossil fuels [4] and are non-biodegradable materials. The gradual exhaustion of conventional fossil fuels makes it significant to develop renewable materials [5,6]. In this context, sound adsorption materials derived from bioresources are attracting increasing attention [7,8,9,10].

Aerogels are porous, lightweight, foam-like materials [11]. Silica aerogel, open-celled mesoporous silica with a high porosity, has been extensively studied because of the unique physicochemical properties, but its practical application is limited due to high mechanical fragility. In recent years, various aerogels derived from abundant biopolymers have been studied, for example, the cellulose aerogel [12,13], chitosan aerogel [14], and lignin aerogel [15] were fabricated. Yet, the acoustic properties of the bio-aerogels have not been fully studied so far. Arabic gum (AG) is an inexpensive polysaccharide, and this biomass is harvested commercially from the *Acacia senegal* and *Acacia seyal* trees, belonging to the Leguminosae family [16]. AG is widely used in the food, cosmetic, and textile industries. The precise structure of AG is little known and it consists of a mixture of arabinogalactan polysaccharide and hydroxyproline protein. The potential of AG in sound adsorption materials has rarely been exploited, and we attempt to use the AG aerogel as a novel sound absorption material in this paper. However, many bio-aerogels (including AG aerogel) are generally plagued with the issue of mechanical fragility; various approaches such as crosslinking [17,18] or filler addition [19,20,21] can be followed to improve aerogel performance [22], and it is highly desirable to tough the bio-aerogel by a simple yet effective method.

Elastomers, more commonly known as rubbers, are widely used as the toughener for polyolefins [23], polylactide [24], epoxy resins [25], etc. To the best of our knowledge, there are few studies that have used rubber as a toughener for aerogel [26]. Herein, we attempt to use the commercially available carboxylic butadiene-acrylonitrite latex (XNBRL) to improve the toughness of the AG bio-aerogel. Considering both XNBRL and AG are both polar materials, they should have good compatibility. The AG/XNBRL hybrid aerogels were prepared by a green method, i.e., the combination of latex compounding and vacuum freeze-drying process. The schematic diagram of the preparation route of AG/XNBRL aerogel is shown in Figure 1. The structure and property of the as-prepared aerogels were characterized by SEM, thermogravimetric analysis, mechanical analysis, and sound adsorption testing. The results revealed that the incorporation of XNBRL could not only improve the toughness, but also improve the thermal stability and sound absorption performances of the resultant aerogels. Overall, our study sheds light on the preparation of mechanical robust bio-based aerogel by a simple method for the sound absorption application.

## 2. Materials and Methods

### 2.1. Materials

Arabic gum (AG, pharmaceutical grade), with a density of 1.35 g/cm^3^, was purchased from Shanghai Aladdin Biochemical Technology Co., Ltd. (Shanghai, China). Carboxylic butadiene-acrylonitrile latex (XNBRL, pH 7–8, solid content 41.5%, viscosity 800–1000 CPS) was kindly provided by Shenzhen Jitian Chemical Co., Ltd. (Shenzhen, China). All other ingredients, such as zinc oxide (ZnO) solution, zinc diethyl dithiocarbamate (ZDEC) solution, tetramethyl thiuram disulfide (TMTD) solution, poly(dicyclopentadiene-p-cresol) (WSL) solution, and sulfur (S) solution were supplied by Guangzhou Polytop Chemical Co., Ltd. (Guangzhou, China).

### 2.2. Preparation of AG/XNBRL Aerogel

Firstly, Arabic gum (AG) powder was dissolved in deionized water under stirring to obtain a 5 wt% transparent solution. Then, a desired amount of carboxylic butadiene-acrylonitrile latex (XNBRL) was added into the Arabic gum aqueous solution under stirring to obtain a uniform AG/XNBRL mixture at room temperature. Subsequently, we added the crosslinking agent sulfur and other rubber additives into the AG/XNBRL mixture solution by violently stirring for 1 h. The dry content of rubber additives is shown as follows: ZnO, 5 phr. Zinc diethyldithiocarbamate, 1.5 phr. Tetramethylthiuram disulfide, 0.5 phr. Sulfur, 2 phr. Antioxidants, 2 phr. Note that “phr” refer to “parts per hundreds of rubber”. After that, the well-mixed solution was vacuumed for one hour at room temperature to eliminate the air bubbles. Then, the mixture solution was subjected to a freeze-drying process for 48 h, and the obtained dried aerogel was subjected to high temperature treatment (120 °C for 30 min) in a vacuum oven to crosslink the rubber phase.

The obtained AG/XNBRL aerogel was coded as A*x*X*y*; A and X refer to AG and XNBRL, *x* and *y* represent the mass ratio for the dried content of AG and dried content of XNBRL. For example, A7X3 means the aerogel consisting of 70 wt% AG and 30 wt% solid content of XNBRL, and A10X0 means the sample consisting of 100 wt% Arabic gum.

### 2.3. Characterization

The true density ($\rho_{t}$) value was obtained by a true density analyzer (3H-2000 TDI, Beijing Beishide Instrument, Beijing, China).

The porosity (P) of aerogel is defined as the percentage of pore volume in the material to the total volume of the material and is calculated as follows [27]:$$P=\left(1-\frac{\rho_{a}}{\rho_{t}}\right)\times 100\%$$
where $\rho_{a}$ and $\rho_{t}$ are the apparent density and true density of the material, respectively.

The scanning electron micrographs (SEM) for the brittled surface of the aerogels (fractured in liquid nitrogen) were observed with a JSM-5510LV microscope instrument (JEOL, Tokyo, Japan). Note that all the samples were spray-coated with a layer of gold before any observations.

Thermogravimetric analysis (TGA) was carried out with a NETZSCH STA409PC comprehensive thermal analyzer at a heating rate of 10 °C∙min^−1^ under N_2_ atmosphere in the temperature range 30–700 °C.

The compressive stress-strain measurements were performed by an electrical universal material testing machine (CMT4104) with a strain rate of 1 mm/min. (compression up to 70% of their original volumes).

Sound absorption performance of the sample was tested by an AWA6128A-type standing wave tube (Beijing Century JT Technology Development Co., Ltd., Beijing, China), the aerogel was cylindrical in shape with a diameter of 30 mm and height of 23 mm.

## 3. Results and Discussion

### 3.1. Thermal Stability

The thermal stability of AG, XNBRL, and AG/XNBRL hybrid aerogels was investigated by thermogravimetric analysis (Figure 2). As shown in Figure 2a, for neat AG, the slight weight loss around 100 °C was ascribed to the release of moisture associated with the hydrophilic character of AG. The AG was quickly decomposed between 230 °C and 335 °C, which is related to the dehydration and decarboxylation reaction of AG [28]. We can also observe the XNBRL were decomposed between 360 °C and 500 °C [29]. Obviously, the neat XNBRL exhibits a better thermal stability compared to neat AG, thereby the T_50%_ for the AG/XNBRL aerogels was gradually improved with the increase of XNBRL content. For example, the T_50%_ was increased from 344 °C for the sample of A7X3 to 432 °C for the sample of A3X7. We can observe that the char residue for AG/XNBRL aerogels decreased with the increase of XNBRL because of the smaller amount of char residue for neat XNBRL compared to neat AG. As shown in Figure 2b, the DTG curves for the AG/XNBRL samples obviously display two peaks, the one occurring at lower temperatures is associated to AG and the second at higher temperatures corresponds to XNBRL. The second peaks became stronger when increasing the XBNRL content.

### 3.2. Mechanical Performance

As is known, mechanical performance is crucial for the real application of aerogel [30], and we therefore investigated the mechanical performances of the AG/XNBRL aerogels. The aspect of the aerogels before compression and their behavior 30 min after compression are shown in Figure 3A. It is observed that the sample made only with XNBRL did not result in aerogel because of serious dimensional shrinkage. In contrast, for the sample of AG and AG/XNBRL aerogels, they all show structural integrality because the rigid AG component could act as skeleton. We also observe that the neat AG aerogels are completely crushed upon compression to 70% of their height, this is because external compression could lead to the inevitable collapse and destruction of the AG aerogels with structural fragility. For the AG/XNBRL aerogels, the external compressive stress could effectively be scattered and dissipated because the crosslinked XNBRL could inhibit/slow crack propagation; it should be noted that the mechanical behavior of the samples should also be correlated with the crosslinking process of the rubber phase. Generally, the crosslinking process of rubber is an indispensable process to make the rubber mechanically robust because the uncross-linked rubber generally exhibits poor mechanical performance. Therefore, the incorporation of crosslinked rubber into the AG aerogel could achieve higher mechanical performances (Figure 3B) [12], for instance, Young’s modulus and 50% modulus (the compress stress at 50% compress strain) are increased from 0.027 MPa and 0.0094 MPa of AG aerogel to 0.272 MPa and 0.4 MPa of A3X7 aerogel, respectively. The compressive stresses of the aerogels increased slowly below 60% strain. After 60% strain, the stresses increased dramatically. This phenomenon was due to the fact that the network skeletons in the aerogels were closely contacted when applied stress induced large strain [31]. The flexibility of the aerogels was further measured by the ratio of shape recovery after compression for various times (Figure 3C). The neat AG aerogel is a typical rigid material in nature, it was crushed into pieces and shows no shape recovery. The shape recoveries of the AG/XNBRL aerogels were significantly improved with the increase of the XNBRL, for example, waiting for 30 min, the sample of A3X7 after compression exhibits a compression behavior similar to that of elastic plastic polymer foam and shows almost 100% shape recovery, attributed to the presence of flexible XNBRL component.

### 3.3. Morphology of Composite Aerogel

The microstructures of the aerogels are shown in Figure 4, all of the aerogels show an orientation that followed the direction of ice crystal growth [32]. The neat AG shows a layered structure, in agreement with a previous report [22], and it is possible to observe that the microstructure of the XNBRL/AG aerogel can be tailored with the increase of XNBRL, i.e., the morphology of XNBRL/AG gradually evolved into a honeycomb-like microstructure. This is most probably due to fact that the XNBRL could combine with the AG with multi-hydrogen bonding and physical entanglement/interlock, thus a porous structure could be clearly identified.

### 3.4. Density and Porosity of Composite Aerogels

The most attractive properties of aerogels are their ultra-low density and high porosity. The apparent density (*ρ_a_*), true density (*ρ_t_*), and porosity of AG/XNBRL aerogels are shown in Table 1. It is observed that the *ρ_a_* are progressively increased from the sample of A10X0 to the sample of A3X7 because of the incorporation of the XNBRL component. Since the density value of AG is higher than for the XNBRL, the higher the ratio of XNBRL:AG, the lower the true density value of the resultant aerogels. It is observed that the porosity of the aerogels decreased upon the XNBRL addition, which is mostly due to the fact that the XNBRL might block the channels or pores to some extent.

### 3.5. Sound Absorption Performance

Porous materials with various channels and cavities are widely used as sound adsorption materials [33]. The sound adsorption performance of AG/XNBRL composites are shown in Figure 5. The sound absorption coefficients for all samples exhibit sharp increase below the frequency of 1500 Hz, and they all show better sound absorption properties at medium-high frequencies compared to low frequencies. With the increase of XNBRL, the average sound adsorption coefficient of the aerogels initially increased and then decreased (Table 2). It is widely accepted that the sound absorption performances are closely related with the structure of channels in the aerogel. The morphologies of the aerogels are gradually involved from layered-structure for neat AG aerogel to honeycomb-like structure for AG/XNBRL aerogel, which could extend the travel path of sound waves and cause multiple reflections and friction of the air flow, leading to a higher energy consumption (Figure 6). Thus, the sample of A7X3 exhibited improved sound-adsorption performances compared to A10X0. Yet, when XNBRL content further increases, the sound-adsorption performances of the aerogels decrease, which is probably due to fact that too much XNBRL may block the channels or pores of the aerogel and therefore weaken the sound adsorption performance of the aerogel. It should be noted that all of the AG/XNBRL aerogels show higher value of average sound adsorption coefficient than the neat AG aerogel (Table 2).

We next use the NRC (noise reduction coefficient) as the indicator to compare the noise absorption ability in our work and other materials reported in previous literature. NRC refers to the arithmetic average of four sound absorption coefficients in the frequencies of 250 Hz, 500 Hz, 1000 Hz, and 2000 Hz. The NRC value in our work reached 0.512, which is better than many commercial insulation materials [34]. Table 3 shows that the as-prepared AG/XNBRL aerogel showed advantages both in sound absorption performance and low density.

## 4. Conclusions

In this research, AG/XNBRL hybrid aerogel structures were prepared for the first time by combining green polysaccharide polymer Arabic gum with carboxylic butadiene-acrylonitrite latex. The AG in the AG/XNBRL aerogel could act as a skeleton to improve the dimensional stability of the aerogel, while the XNBRL phase could effectively improve the toughness of the ensuing aerogel. The morphologies of the aerogels are involved from layered-structure for neat AG aerogel to honeycomb-like structure for AG/XNBRL aerogel, which could extend the travel path of sound waves and cause multiple reflections and friction of the air flow, leading to an enhanced sound adsorption performance. The as-prepared AG/XNBRL hybrid aerogel exhibits excellent toughness, thermal stability, and sound absorption properties. Overall, our study sheds light on a promising design strategy of flexible bio-based aerogel for sound absorption application.

## Figures and Tables

**Figure 1 polymers-14-01344-f001:**
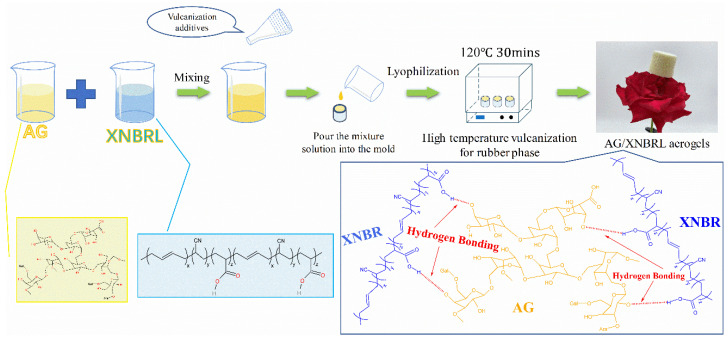
Schematic diagram of the preparation route for AG/XNBRL aerogel.

**Figure 2 polymers-14-01344-f002:**
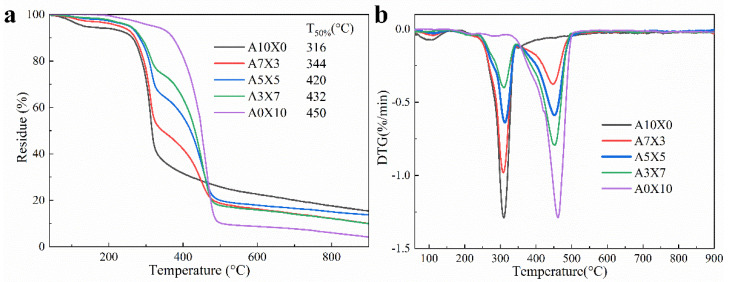
(**a**) TGA and (**b**) DTG curves of AG/XNBRL composite aerogels with different ratios. T_50%_ refers to 50% weight loss temperature.

**Figure 3 polymers-14-01344-f003:**
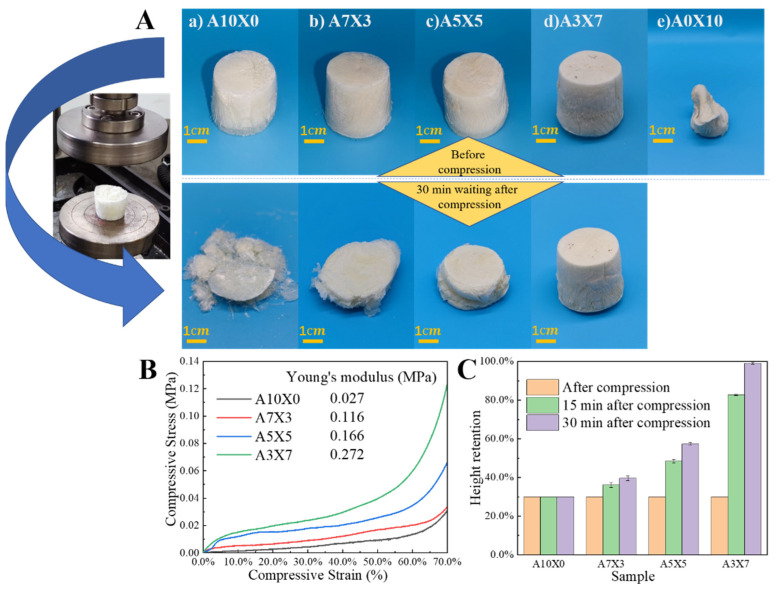
Mechanical performances for the AG/XNBRL aerogels. (**A**) Photos for AG/XNBRL aerogels before compression (upper panel) and 30 min waiting after compression (lower panel). (**B**) Compression curves of AG/XNBRL aerogels. (**C**) Height recovery of the samples after compression with various waiting times.

**Figure 4 polymers-14-01344-f004:**
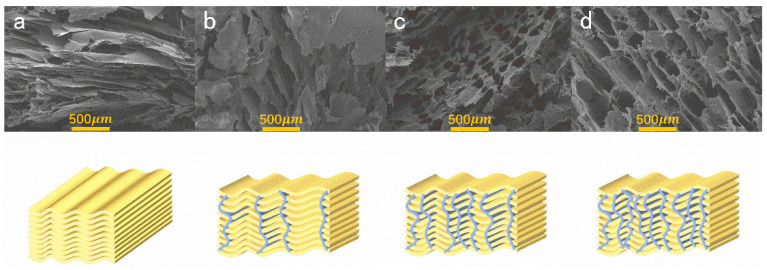
SEM photos and schematic diagrams of AG/XNBRL aerogels: (**a**)A10X0, (**b**)A7X3, (**c**) A5X5, (**d**) A3X7.

**Figure 5 polymers-14-01344-f005:**
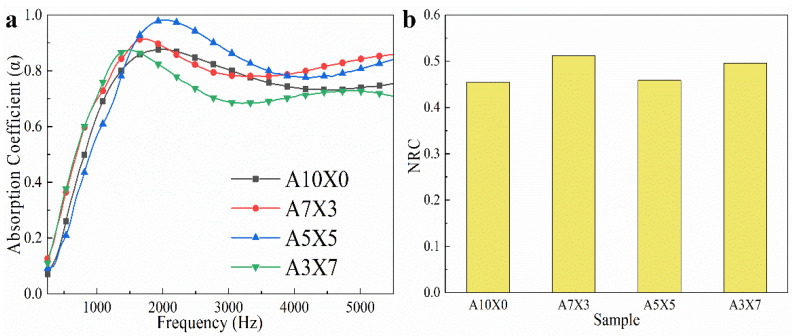
(**a**) The curves of sound absorption coefficient and (**b**) the NRC value for various aerogels.

**Figure 6 polymers-14-01344-f006:**
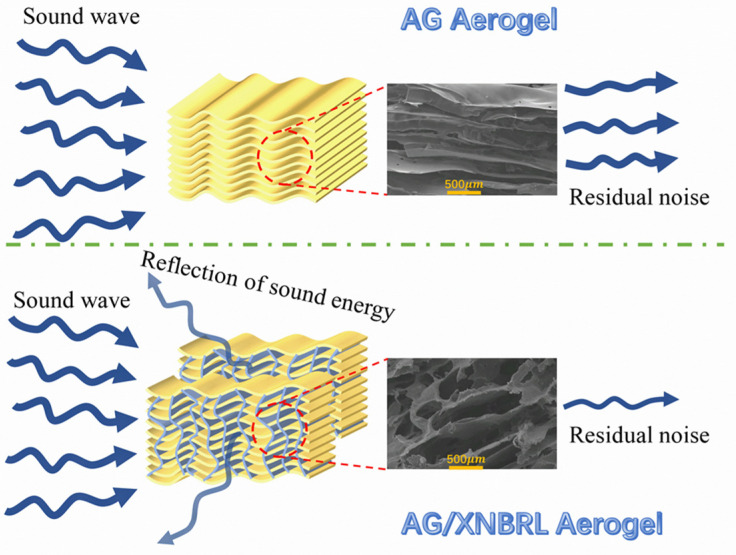
Sound absorption mechanism of AG aerogels and AG/XNBRL composite aerogels.

**Table 1 polymers-14-01344-t001:** The apparent density (*ρ_a_*), true density (*ρ_t_*), and porosity of AG/XNBRL aerogels.

Samples	*ρ_a_* (g/cm^3^)	*ρ_t_* (g/cm^3^)	Porosity (%)
A10X0	0.0508	1.36	96.3
A7X3	0.0692	1.22	94.3
A5X5	0.092	1.13	91.9
A3X7	0.134	1.03	87

**Table 2 polymers-14-01344-t002:** The average sound adsorption coefficient of the aerogels.

Sample	250 Hz	500 Hz	1000 Hz	2000 Hz	3000 Hz	4000 Hz	5000 Hz	Average
A10X0	0.070	0.230	0.639	0.877	0.804	0.739	0.738	0.585
A7X3	0.126	0.338	0.697	0.887	0.784	0.791	0.842	0.638
A5X5	0.091	0.194	0.567	0.981	0.869	0.779	0.807	0.613
A3X7	0.110	0.353	0.707	0.813	0.688	0.707	0.727	0.586

**Table 3 polymers-14-01344-t003:** Sound absorption properties of various materials [35,36,37].

Materials	Thickness (mm)	Density (g/cm^3^)	NRC
Our work	23	0.060	0.512
Aerated concrete	90	0.670	0.165
Cane board	13	0.200	0.375
Superfine glass wool	20	0.020	0.425
Cement expanded perlite slab	80	0.300	0.430
Microcellular polyurethane foam	40	0.030	0.430
Coarse-porous polyurethane foam	40	0.030	0.443
Phenolic resin glass wool board	30	0.100	0.465
Perlite suction panel	18	0.340	0.343
Nitrocellulose foam	25	0.025	0.458
Urea-formaldehyde miboro	30	0.020	0.485
Concrete with lightweight aggregate	25	2.310	0.150
Geopolymer concrete with lightweight aggregate	<10	1.510	0.290
Plaster with lightweight aggregate	10	0.300	0.060
Alkali-activated cellular concrete	-	0.720	0.410
Pervious concrete	4	0.640	0.350
Hemp concrete	5	0.590	0.450
Metal fiber porous materials	2.1	-	0.044
β-HPG porous sound-absorbing material	40	-	0.320

## Data Availability

Not applicable.

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
