# Peer review of "Enhanced Toughness and Sound Absorption Performance of Bio-Aerogel via Incorporation of Elastomer"

_polymers, 2022, doi:10.3390/polym14071344_

Round 1
Reviewer 1 Report
- What it means: solution by violently stirring for 1h (Raw 84) explain?!
The manuscript must clearly present the Preparation of AG/XNBRL aerogel, namely sample notation with the correspondingly compounds and ratio between them;
- Authors must include as well the thermal stability of AG and XNBRL for comparison and conclusions about thermal stability of the blends;
- Raw 138: We next to study the mechanical performance of the aerogels as the mechanical performances of aerogels are crucial for many applications ? !
- The aspect of the aerogels before compression and their behaviour 30 min after compression are shown in Figure 3A; it is observed that the sample made only with XNBR did not result in aerogel because of serious dimensional shrinkage. …..instead of The aerogels before compression and 30 min waiting after compression are shown in Figure 3A, it is observed that the sample made up only with XNBR did not result in aerogel because of serious dimensional shrinkage.
- the authors claim: For the AG/XNBR aerogels, the external compressive stress could effectively be scattered and dissipated because the XNBR could act as buffer substance to inhibit/slow crack propagation, therefore resulting in higher mechanical performances. What it means in their oppinion buffer substance? on the other hand in the experimental part the crosslinking process is specified; The mechanical behavior of the samples should probably be correlated with this process
- Figure 4. SEM photos and schematic diagrams of AG/XNBRL aerogels: (a)A10X0 163 (b)A7X3 (c) A5X5 (d) A3X7. – These are not presented
- the manuscript needs substantial improvements to be published
Author Response
|
Title of submitted paper : |
Enhanced toughness and sound absorption performance of bio-aerogel via incorporation of elastomer |
|
Authors : |
Junshi Shen , Ruofei Hu * , Xueliang Jiang * , Feng You , Chu Yao , Huan Yang , Peng Yu * |
|
Manuscript ID |
polymers-1639978 |
We thank the Reviewers and Editors for their useful comments. The manuscript has been modified according to these comments.
Reviewer(s)' Comments to Author:
Comments:
Point 1: What it means: solution by violently stirring for 1h (Raw 84) explain?! The manuscript must clearly present the Preparation of AG/XNBRL aerogel, namely sample notation with the correspondingly compounds and ratio between them;
Response 1: Thanks for the valuable suggestion, and we made the revision. We added the crosslinking agent sulphur and other rubber additives into the AG/XNBRL mixture solution by violently stirring for 1h.
The detailed procedures and sample notation are shown as follows:
We firstly prepared the 5 wt% arabic gum (AG) aqueous solution, then, a desired amount of carboxylic butadiene-acrylonitrite latex (XNBRL) was added into the arabic gum aqueous solution under stirring to obtain a uniform AG/XNBRL mixture at room temperature. We next added the crosslinking agent sulphur and other rubber additives into the AG/XNBRL mixture solution by violently stirring for 1h. After that, the mixture solution was vacuumed, freeze-dried, and cured successively to obtain the AG/XNBRL aerogel.
The obtained AG/XNBRL aerogel was code as AxXy, A and X refers to AG and XNBRL, x and y represents the mass ratio for the dried content of AG and dried content of XNBRL, for example, A7X3 means the aerogel consisting of 70 wt% AG and 30 wt% solid content of XNBRL, and the A10X0 means the sample consisting of 100 wt% AG.
Point 2: Authors must include as well the thermal stability of AG and XNBRL for comparison and conclusions about thermal stability of the blends;
Response 2: Thanks for the valuable suggestion, and we made the revision.
Figure. (a) TGA and (b) DTG curves of AG/XNBRL composite aerogels with different ratios. T50% refers to 50% weight loss temperature.
The thermal stability of AG, XNBRL and AG/XNBRL hybrid aerogels was investigated by thermogravimetric analysis (Figure). For neat AG, the slight weight loss around 100 °C was ascribed to the release of moisture associated with the hydrophilic character of AG. AG was quickly decomposed between 230 °C and 335 °C, which is related to the dehydration and decarboxylation reaction of AG [1]. We can also observe the XNBRL were decomposed between 360 °C and 500 °C [2]. Obviously, the neat XNBRL exhibit a better thermal stability compared to neat AG, thereby the T50% for the AG/XNBRL aerogels was gradually improved with the increase of XNBRL content. For example, the T50% was increased from 344 °C for the sample of A7X3 to 432 °C for the sample of A3X7. Also, we can observe that the char residue for AG/XNBRL aerogels decreased with the increase of XNBRL because of the less char residue for neat XNBRL compared to neat AG. As shown in Fig b, the DTG curves for the AG/XNBRL samples obviously display two peaks, the one occurring at lower temperatures is associated to AG and the second at higher temperatures corresponds to XNBRL. The second peaks became stronger when increasing the XBNRL content.
The cited references are shown as follows.
- Cozic C, Picton L, Garda M R, et al. Analysis of arabic gum: Study of degradation and water desorption processes[J]. Food Hydrocolloids, 2009, 23(7): 1930-1934.
- Yu, Z., Li, J., Yang, L., Yao, Y., Su, Z., & Chen, X. (2012). Synthesis and properties of nano carboxylic acrylonitrile butadiene rubber latex toughened phenolic resin. Journal of Applied Polymer Science, 123(2), 1079-1084.
Point 3: Raw 138: We next to study the mechanical performance of the aerogels as the mechanical performances of aerogels are crucial for many applications?!
Response 3: Revision
“We next to study the mechanical performance of the aerogels as the mechanical performances of aerogels are crucial for many applications” is changed to “As is known, the mechanical performances is crucial for the real application of aerogel, and we therefore investigated the mechanical performances of the AG/XNBRL aerogels”
Point 4: The aspect of the aerogels before compression and their behaviour 30 min after compression are shown in Figure 3A; it is observed that the sample made only with XNBRL did not result in aerogel because of serious dimensional shrinkage. …..instead of The aerogels before compression and 30 min waiting after compression are shown in Figure 3A, it is observed that the sample made up only with XNBRL did not result in aerogel because of serious dimensional shrinkage.
Response 4: Thanks for the reviewer for the valuable suggestion, and we make the revision as the reviewer suggested.
Point 5: the authors claim: For the AG/XNBRL aerogels, the external compressive stress could effectively be scattered and dissipated because the XNBRL could act as buffer substance to inhibit/slow crack propagation, therefore resulting in higher mechanical performances. What it means in their opinion buffer substance? on the other hand in the experimental part the crosslinking process is specified; The mechanical behavior of the samples should probably be correlated with this process
Response 5: Thanks for the reviewer for the valuable suggestion, the “buffer substance” refers to the “XNBRL phase”. It is quite true that the mechanical behavior of AG/XNBRL aerogel is closely related to the crosslinked XNBRL phase. Generally, the crosslinking process of rubber is an indispensable process to make the rubber mechanically robust because the uncross linked rubber generally exhibits poor mechanical performance.
“For the AG/XNBRL aerogels, the external compressive stress could effectively be scattered and dissipated because the XNBRL could act as buffer substance to inhibit/slow crack propagation, therefore resulting in higher mechanical performances.” is changed as “For the AG/XNBRL aerogels, the external compressive stress could effectively be scattered and dissipated because the crosslinked XNBRL could inhibit/slow crack propagation, it should be noted that the mechanical behavior of the samples should also be correlated with the crosslinking process of the rubber phase. Generally, the crosslinking process of rubber is an indispensable process to make the rubber mechanically robust because the uncross linked rubber generally exhibits poor mechanical performance. Therefore, the incorporation of crosslinked rubber into the AG aerogel could achieve higher mechanical performances.”
Point 6: Figure 4. SEM photos and schematic diagrams of AG/XNBRL aerogels: (a)A10X0 163 (b)A7X3 (c) A5X5 (d) A3X7. – These are not presented
Response 6: We added the figure in the revised manuscript.
Figure 4. SEM photos and schematic diagrams of AG/XNBRL aerogels: (a)A10X0 (b)A7X3 (c) A5X5 (d) A3X7.
Point 7: the manuscript needs substantial improvements to be published
Response 7: We tried our best to improve the manuscript.
We thank the Editor and Reviewers for their helpful comments on this paper!!!

Reviewer 2 Report
Revision for Polymers (ISSN 2073-4360)
Manuscript ID: polymers-1639978
Title: Enhanced toughness and sound absorption performance of bio-aerogel via incorporation of elastomer
List of authors: Junshi Shen , Ruofei Hu * , Xueliang Jiang * , Feng You , Chu Yao , Huan Yang , Peng Yu *
This research paper deals with the preparation of arabic gum/carboxybutyronitrile rubber aerogels (AG/XNBRL) hybrid aerogel by a green methodology. The hybrid material showed a good combination of properties derived from its components. AG/XNBRL hybrid exhibited boosted toughness compared to AG, but also improved thermal stability and sound absorption performances. I consider the research work a valuable activity and it paves the way for future prospectives in the field of noise pollution reduction and photoabsorbance. As the subject is interesting, I am willing to recommend minor revisions with pending manuscript decision. I will gladly be able to review the modified manuscript once the following points have been fully addressed:
1) Page 1, lines 20-22 – Could the authors quantify the improved thermal stability and sound absorption reporting some numbers? For example, the maximum sound absorption coefficient.
2) Page 1, lines 36-37 – At this point, I would suggest to add some references reporting the use of bio-compatible materials for the manufacturing of sound absorption composites. I would recommend integrating the references with the following to support the direction of scientific community toward the development of renewable materials:
- Khan, W. S., Asmatulu, R., & Yildirim, M. B. (2012). Acoustical properties of electrospun fibers for aircraft interior noise reduction. Journal of Aerospace Engineering, 25(3), 376-382.
- Hajimohammadi, M., Soltani, P., Semnani, D., Taban, E., & Fashandi, H. (2022). Nonwoven fabric coated with core-shell and hollow nanofiber membranes for efficient sound absorption in buildings. Building and Environment, 108887.
- Xu, F., Zhang, S., Wang, G., Zhao, D., Feng, J., Wang, B., & He, X. (2021). Lightweight Low‐Frequency Sound‐Absorbing Composites of Graphene Network Reinforced by Honeycomb Structure. Advanced Materials Interfaces, 8(16), 2100183.
- Passaro, J., Russo, P., Bifulco, A., De Martino, M. T., Granata, V., Vitolo, B., ... & Branda, F. (2019). Water resistant self-extinguishing low frequency soundproofing polyvinylpyrrolidone based electrospun blankets. Polymers, 11(7), 1205.3) Page 3, lines 141-144 - I would cite more useful references to support these statements.
4) Page 2, lines 52-53 – I would cite more references at this point.
5) Page 3, lines 103-104 – I would suggest to add a reference to support the formula used to calculate the porosity “P”.
6) Page 4, line 124 – Replace “we” with capital letter “We”.
7) Page 4, lines 129-131 – I would suggest to add TGA derivative curves (DTG) for showing the decomposition peaks and better support the thermal stability discussion.
8) Page 5, line 167 – Replace “we can observed” with it is possible to observe”.

Author Response
|
Title of submitted paper : |
Enhanced toughness and sound absorption performance of bio-aerogel via incorporation of elastomer |
|
Authors : |
Junshi Shen , Ruofei Hu * , Xueliang Jiang * , Feng You , Chu Yao , Huan Yang , Peng Yu * |
|
Manuscript ID |
polymers-1639978 |
We thank the Reviewers and Editors for their useful comments. The manuscript has been modified according to these comments.
Reviewer(s)' Comments to Author:
Comments:
This research paper deals with the preparation of arabic gum/carboxybutyronitrile rubber aerogels (AG/XNBRL) hybrid aerogel by a green methodology. The hybrid material showed a good combination of properties derived from its components. AG/XNBRL hybrid exhibited boosted toughness compared to AG, but also improved thermal stability and sound absorption performances. I consider the research work a valuable activity and it paves the way for future prospectives in the field of noise pollution reduction and photoabsorbance. As the subject is interesting, I am willing to recommend minor revisions with pending manuscript decision. I will gladly be able to review the modified manuscript once the following points have been fully addressed:
Point 1: Page 1, lines 20-22 – Could the authors quantify the improved thermal stability and sound absorption reporting some numbers? For example, the maximum sound absorption coefficient.
Response 1: Thanks for the reviewer for the valuable suggestion, revision.
“for instance, the half weight loss (50%) temperature and average sound adsorption coefficient for the aerogel containing 30wt% XNBRL is 344 °C and 0.585, respectively, which are superior than those of neat AG aerogel.” is added in the revised manuscript.”
Point 2: Page 1, lines 36-37 – At this point, I would suggest to add some references reporting the use of bio-compatible materials for the manufacturing of sound absorption composites. I would recommend integrating the references with the following to support the direction of scientific community toward the development of renewable materials:
- Khan, W. S., Asmatulu, R., & Yildirim, M. B. (2012). Acoustical properties of electrospun fibers for aircraft interior noise reduction. Journal of Aerospace Engineering, 25(3), 376-382.
- Hajimohammadi, M., Soltani, P., Semnani, D., Taban, E., & Fashandi, H. (2022). Nonwoven fabric coated with core-shell and hollow nanofiber membranes for efficient sound absorption in buildings. Building and Environment, 108887.
- Xu, F., Zhang, S., Wang, G., Zhao, D., Feng, J., Wang, B., & He, X. (2021). Lightweight Low‐Frequency Sound‐Absorbing Composites of Graphene Network Reinforced by Honeycomb Structure. Advanced Materials Interfaces, 8(16), 2100183.
- Passaro, J., Russo, P., Bifulco, A., De Martino, M. T., Granata, V., Vitolo, B., ... & Branda, F. (2019). Water resistant self-extinguishing low frequency soundproofing polyvinylpyrrolidone based electrospun blankets. Polymers, 11(7), 1205.3) Page 3, lines 141-144 - I would cite more useful references to support these statements.
Response 2: Thanks for the suggesting of the valuable references, these references could significantly improve the quality of the manuscript, we cited these references in the revised manuscript.
- Khan, W. S., Asmatulu, R., & Yildirim, M. B. (2012). Acoustical properties of electrospun fibers for aircraft interior noise reduction. Journal of Aerospace Engineering, 25(3), 376-382.
- Hajimohammadi, M., Soltani, P., Semnani, D., Taban, E., & Fashandi, H. (2022). Nonwoven fabric coated with core-shell and hollow nanofiber membranes for efficient sound absorption in buildings. Building and Environment, 108887.
- Xu, F., Zhang, S., Wang, G., Zhao, D., Feng, J., Wang, B., & He, X. (2021). Lightweight Low‐Frequency Sound‐Absorbing Composites of Graphene Network Reinforced by Honeycomb Structure. Advanced Materials Interfaces, 8(16), 2100183.
- Passaro, J., Russo, P., Bifulco, A., De Martino, M. T., Granata, V., Vitolo, B., ... & Branda, F. (2019). Water resistant self-extinguishing low frequency soundproofing polyvinylpyrrolidone based electrospun blankets. Polymers, 11(7), 1205.
Point 4: Page 2, lines 52-53 – I would cite more references at this point.
Response 4: We cited more references in the revised manuscript. The cited references are shown as follows.
Crosslinking:
- Khedaioui, D., Boisson, C., d'Agosto, F., & Montarnal, D. (2019). Polyethylene Aerogels with Combined Physical and Chemical Crosslinking: Improved Mechanical Resilience and Shape‐Memory Properties. Angewandte Chemie International Edition, 58(44), 15883-15889.
- Zhu, G., Chen, Z., Wu, B., & Lin, N. (2019). Dual-enhancement effect of electrostatic adsorption and chemical crosslinking for nanocellulose-based aerogels. Industrial Crops and Products, 139, 111580.
Filler addition:
- Pan, J., Li, Y., Chen, K., Zhang, Y., & Zhang, H. (2021). Enhanced physical and antimicrobial properties of alginate/chitosan composite aerogels based on electrostatic interactions and noncovalent crosslinking. Carbohydrate Polymers, 266, 118102.
- Karamikamkar, S., Naguib, H. E., & Park, C. B. (2020). Advances in precursor system for silica-based aerogel production toward improved mechanical properties, customized morphology, and multifunctionality: A review. Advances in colloid and interface science, 276, 102101.
Both:
- de Luna, M. S., Ascione, C., Santillo, C., Verdolotti, L., Lavorgna, M., Buonocore, G. G., ... & Ambrosio, L. (2019). Optimization of dye adsorption capacity and mechanical strength of chitosan aerogels through crosslinking strategy and graphene oxide addition. Carbohydrate polymers, 211, 195-203.
Point 5: Page 3, lines 103-104 – I would suggest to add a reference to support the formula used to calculate the porosity “P”.
Response 5: It is quite true that we should add more references to support the formula used to calculate the porosity “P”.
The cited reference is shown as follows.
- Voronina, N. (1997). An empirical model for rigid frame porous materials with high porosity. Applied Acoustics, 51(2), 181-198.
Point 6: Page 4, line 124 – Replace “we” with capital letter “We”.
Response 6: Thanks for the valuable suggestion, we apologized for the mistake, and we made the revision as the reviewer suggested.
Point 7: Page 4, lines 129-131 – I would suggest to add TGA derivative curves (DTG) for showing the decomposition peaks and better support the thermal stability discussion.
Response 7: Thanks for the valuable suggestion, the DTG curves is added in the revised manuscript.
Figure. (a) TGA and (b) DTG curves of AG/XNBRL composite aerogels with different ratios. T50% refers to 50% weight loss temperature.
As shown in Fig b, the DTG curves for the AG/XNBRL samples obviously display two peaks, the one occurring at lower temperatures is associated to AG and the second at higher temperatures corresponds to XNBRL. The second peaks became stronger when increasing the XBNRL content.
The cited references are shown as follows.
- Cozic C, Picton L, Garda M R, et al. Analysis of arabic gum: Study of degradation and water desorption processes[J]. Food Hydrocolloids, 2009, 23(7): 1930-1934.
- Yu, Z., Li, J., Yang, L., Yao, Y., Su, Z., & Chen, X. (2012). Synthesis and properties of nano carboxylic acrylonitrile butadiene rubber latex toughened phenolic resin. Journal of Applied Polymer Science, 123(2), 1079-1084.
Point 8: Page 5, line 167 – Replace “we can observed” with it is possible to observe”.
Response 8: Thanks for the valuable suggestion, and we made the revision as the reviewer suggested.
We thank the Editor and Reviewers for their helpful comments on this paper!!!

Round 2
Reviewer 1 Report
the manuscript still contains incomplete sentence as for example:
mixture solution by violently stirring for 1h? how many rotation/min for example;
anyway the authors generally answered to solicited corrections